# Options for Fertility Treatments for Trans Women in Germany

**DOI:** 10.3390/jcm8050730

**Published:** 2019-05-22

**Authors:** Florian Schneider, Bettina Scheffer, Jennifer Dabel, Laura Heckmann, Stefan Schlatt, Sabine Kliesch, Nina Neuhaus

**Affiliations:** 1Institute of Reproductive and Regenerative Medicine, Centre of Reproductive Medicine and Andrology, Albert-Schweitzer Campus 1, Building D11, 48149 Muenster, Germany; florian.jo.schneider@googlemail.com (F.S.); bettina.scheffer@ukmuenster.de (B.S.); jennifer.dabel@ukmuenster.de (J.D.); Laura.heckmann@ukmuenster.de (L.H.); nina.neuhaus@ukmuenster.de (N.N.); 2Department of Clinical and Surgical Andrology, Centre of Reproductive Medicine and Andrology, Albert-Schweitzer Campus 1, Building D11, 48149 Muenster, Germany; sabine.kliesch@ukmuenster.de

**Keywords:** testes, transgender, fertility preservation, gender confirming hormone therapy, gender confirming surgery, sperm, cryopreservation, trans women

## Abstract

Fertility preservation in trans women is a crucial but thus far neglected component in the gender confirming treatment in Germany. It is difficult for trans women to access reproductive health care because centers offering treatment, psychological guidance, gender confirming surgery, as well as reproductive health services are scarce in Germany. Legal, social, or financial issues as well as individual patient comorbidities prevent trans women from receiving appropriate counselling. This review provides an overview on options of fertility preservation in trans women. We consider recent publications on testicular regression at the time of gender confirming surgery demonstrating presence of sperm or at least spermatogonia in the majority of tissues. This may open options for cryopreservation of sperm or testicular stem cells in trans women even at the final stage of transition. Hence, standardized urological procedures (i.e., sperm cryopreservation after masturbation or sperm extraction from the testicular tissue) and experimental approaches (cryopreservation of testicular tissue with undifferentiated spermatogonia) can be offered best at the initiation but also during the gender confirming process. However, counselling early in the gender confirming process increases the chances of fertility preservation because gender confirming hormone therapy has an impact on spermatogenesis.

## 1. Introduction

Fertility preservation is offered to patients at risk of becoming infertile [1]. The hormonal as well as surgical treatment of transgender patients results in infertility. The German and international guidelines stress that counselling about fertility options and fertility protection is an essential part of the treatment of transgender patients. The standards of care of the World Professional Association of Transgender Health (WPATH) state that counselling about fertility options during the gender confirming procedures is mandatory and an integral part of care for transgender [2]. Treatment of transgender is individualized and performed by a multi-disciplinary team of clinicians. The German guidelines for the treatment of transgender patients were recently published [3]. In a concerted action endocrinologists, general practitioners, gynecologists, urologists, psychotherapists, and reproductive physicians take part in the care during the gender confirming process. Following extensive counseling on available options, the individual patient can decide if any and which technique of fertility preservation will be performed [4]. Limited information is available on the welfare of children born to transgender persons and conceived via assisted reproduction [5]. However, a recent study reported no pathologies in their well-being or psychological development and no difficulties in their relationships with their parents [6]. Therefore, there is no justification to exclude transgender patients from options of assisted reproduction with regard to the well-being of their children [7]. Transgender patients may have biological children and treatment guidelines command the clinicians to offer the most appropriate treatment.

While clinical guidelines request the clinician to inform the transgender patient about fertility options, transgender patients themselves often ask for treatment advice. The majority of transgender expressed interest to learn about fertility preservation during the gender confirming process [8]. Reports of transgender individuals seeking counselling is increasing worldwide [9,10,11,12,13], however, the proportion of patients of the published cohorts taking measures to preserve fertility is still low. Moreover, interest in fertility preservation among transgender individuals may highly vary in different countries [14]. The majority of patients of a Belgian study would have considered sperm freezing if it had been presented as an option [8] and only a small percentage thought that sperm freezing was in conflict with their female identity [15]. The majority (70%) of German transgender patients wish to have children [16]. In Germany, two out of 99 trans women and three out of 90 trans men had their own children, but four of these five individuals stated not to have used their own oocytes or sperm [16]. Rather, 16% of trans men had biological children with donor sperm [17]. Apart from that, 88% of trans men used their own oocytes and mean age at conception was 28 [18]. Those participants judging genetic parenthood as important were more prone to undertake steps for fertility preservation [14]. A majority of over 1000 random participants (76.2%) indicated that doctors should enable transgender individuals to have their own children [19].

Do individuals undergoing gender confirming hormone therapy (GCHT) still have options for fertility preservation? GCHT in trans women consists of treatment with anti-androgens (cyproterone acetate) or spironolactone or gonadotropin releasing hormone agonists in combination with estrogens (oral, transdermal, intramuscular). This leads to down-regulation of gonadotropin releasing hormone, followed by a prominent decline of luteinizing hormone (LH) and follicle stimulating hormone (FSH). This leads to testicular regression associated with low levels of serum androgens and suppression of spermatogenesis. After several weeks GCHT interrupts germ cell differentiation at the pre-meiotic stage [20,21,22]. Byne et al. concluded that counselling with regard to fertility protection/preservation should therefore start prior to initiation of GCHT [23]. The efficiency of fertility preservation depends on the quality and quantity of sperm which can be cryopreserved [24]. Effects of GCHT for trans women may not always be reversible once initiated [25]. Different standardized andrological/urological procedures including cryopreservation of sperm from the ejaculate, microsurgical testicular sperm extraction (mTESE) or microsurgical epididymal sperm aspiration (MESA) can be offered during the gender confirming process in order to preserve fertility in trans women. Counselling as early as possible in the gender confirming process is crucial as the testes are still intact providing the chance to cryopreserve sperm from the ejaculate. Cryopreservation of spermatogonial stem cells (SSCs) is also possible, however, using them for fertility procedures is not yet an established clinical routine but is still in the experimental phase.

A total of 90% of transgender persons stated that fertility loss was not relevant to defer the decision to engage in gender affirming medical interventions. [8]. Many trans women favor a fast transition over fertility concerns [5]. The desire of a fast transition and the potentially poor quality of spermatogenesis urge the clinicians to address fertility options as soon as possible at the initial stages of the gender confirming process. Discussing fertility protection early, even in younger patients, might increase the chance that transgender patients approach fertility centers. Wakefield et al. showed a considerably higher rate of concern for fertility issues in younger people in comparison to adults (75% vs. 13.5%) before initiation of treatment [26]. In the Netherlands, 91% of trans girls were counselled for fertility protection and most of them (75%) were able to cryopreserve sperm [27]. Surprisingly, only a minority felt uncomfortable with masturbation (17%) for generating a sperm sample. With regard to the parents of adolescent transgender patients, 65% felt the desire to know the impact of GCHT on fertility and wanted their children to consider fertility preservation methods [28]. These studies demonstrate that early counselling raises the concern and improves the chances of creating a fertility reserve.

As yet, no transgender clinic in Germany combines all specialists in one center. This may be desirable due to the significant number of patients with an incidence of 6.8 trans women in 100,000 people worldwide [29]. Several publications confirmed the need for counselling regarding fertility preservation by experienced medical staff during the transition process. However, state laws, medical regulations, financial restrictions, social stigmas, and scarcity of experienced centers hinder a more widespread application of fertility protection in transgender patients [25,30,31,32]. Albeit reproductive health staff expresses a positive attitude towards transgender patients, they often have to adjust and adopt appropriate ways of dealing with transgender patients [33]. Healthcare professionals can benefit, however, from knowledge gained through communication with patients about their desires to have children [34].

## 2. Testicular Tissue and Steroidogenesis of Trans Women

The human testis consists of the interstitium and the tubular compartment (seminiferous tubules). In the interstitium Leydig cells produce testosterone (steroidogenesis). This compartment is also populated by immune and neural cells as well as blood and lymphatic vessels. The tubular compartment consists of peritubular cells, Sertoli cells and germ cells. SSCs are located at the basement membrane of the seminiferous tubules and are capable of undergoing self-renewal as well as differentiating cell divisions. When spermatogonia begin to differentiate they form clones by mitotic divisions. They then enter into meiosis to become spermatocytes. After the two meiotic divisions they populate the seminiferous epithelium as spermatids. After spermiogenesis they are released into the lumen as immotile spermatozoa. Steroidogenesis and spermatogenesis are not only located in separate compartments, they are also controlled by different partly interrogating hormonal axes (LH-testosterone; FSH-inhibin) [35]. As mentioned earlier, GCHT suppresses steroidogenesis at the hypothalamic level during the gender confirming process creating via exogenous hormonal regimens feminized blood hormone levels (low serum testosterone, high serum estradiol) and inhibition of spermatogenesis.

## 3. Testicular Tissues and Steroidogenesis in Our Cohort of Trans Women

As published in 2015, we collected in a multi-center setting questionnaires, testicular tissues, and blood serum to evaluate the level of spermatogenesis and steroidogenesis of each trans woman on the day of gender confirming surgery (GCS) [20]. Up to today (between February 2012 and June 2018) we evaluated testes from 238 individuals (354 testicular tissues) and 226 questionnaires of 268 trans women. Ethical approval was received from the Ethical Committee of the Ärztekammer Westfalen-Lippe (no. 2012-555-f-S) and the Ärztekammer Hamburg (no. MC-131/13) prior to study initiation and written informed consent was obtained from each subject prior to study participation. Patients were recruited from three different German clinics applying diverse GCHT protocols. Of these, 48 persons took GCHT until GCS, 178 stopped the therapy two weeks before, and another 40 persons 4–6 weeks before GCS. The mean age was 41 years with the youngest patient being 16 and the oldest being 66 years of age. Most patients were of reproductive age. Moreover, 168 patients received cyproterone acetate and estrogens as GCHT. The individual treatment differed, furthermore, in the dose regimens. The minimal dose was 0.25 mg/d and the maximal dose was 150 mg/d with the average dose of cyperoterone acetate being 23.63 mg/d. The mean period of GCHT was 29.15 months with a shortest duration of five months and the longest duration of 169 months prior GCS. Some patients were operated too early which was not in full accordance with national and international guidelines. Long waiting lists for surgeries, complications with endocrine treatments, and psychological problems were reasons why some individuals showed extended GCHT until GCS. Mean serum testosterone was 8.9 nmol/L (SD 8.6 nmol/L) and mean serum estradiol was 195.9 pmol/L (SD 382.4 pmol/L) on the day of GCS.

There were 13 publications that reported histological findings of testicular tissues of trans women (Figure 1). The results varied from spermatogenic suppression with a hyalinized basal membrane and a decreased Leydig cell function to a heterogeneous picture regarding the spermatogenic state (Figure 2) [36]. In a recent US publication examining 99 transgender testes, a decreased diameter of seminiferous tubules and expansion of the interstitium, marked hypoplasia of germ cells, hypoplasia or absence of Leydig cells, and epididymal hyperplasia were found [21]. Another publication from Thailand examining 173 testicular tissues stated that feminizing hormonal treatment before GCS results in abnormalities of germ cell development and loss of reproductive functions [22]. Unfortunately, in both studies, endocrine profiles were not determined.

We confirmed a highly heterogeneous histological appearance after examining testicular tissues from 108 patients in 2015. Among the patients, 24% had complete spermatogenesis on the day of GCS [20]. Evaluating spermatogenesis in a larger group of transgender patients (238 testes, Figure 2) over six years up to today, we evaluated 39 testicular tissues with complete spermatogenesis, 63 with meiotic arrest, 95 with spermatogonial arrest, and 41 with Sertoli cell only syndrome (SCO) or tubular shadows applying a systematic scoring scheme [47]. In the publication of Matoso et al., out of 99 testicular tissues, 80% showed spermatogenic defects with maturation arrest at the spermatogonial stage [21]. The remaining 20% had marked hypospermatogenesis with few and focal spermatids. In the publication from Jindarak et al. [22], 173 testicular tissues were examined: 11% had normal spermatogenesis, 26% showed hypospermatogenesis, 36.4% showed maturation arrest, 20.2% Sertoli cell only syndrome, and 6.4% seminiferous tubule hyalinization [22]. All publications show severe inhibition of spermatogenesis. Strikingly, the presence of focal areas of spermatogenesis in testes of trans women at the time of GCS generates an opportunity to extract spermatids from these areas. Alternatively, spermatogonial stem cells can be cryopreserved in the few cases of complete germ cell arrest (Figure 3). To assess the characteristics of cell types in testicular tissues of trans women, immunohistochemical stainings were performed. The use of immunohistochemical markers (lutropin-choriogonadotropic hormone receptor (LHCGR), smooth muscle actin (SMA), vimentin (VIM), sal like protein 4 (SALL 4), melanoma-associated antigen 4 (MAGE A4), DEAD-box helicase 4 DDX4/VASA) revealed that the expression patterns in trans women were identical and in agreement with the staining pattern in samples from non-transgender men [48]. This finding was confirmed by others [21,22].

In general, spermatogenesis is suppressed during GCHT, but complete spermatogenesis or tubules with sperm or spermatozoa are still present in tissues even on the day of GCS, which could help the reproductive clinician to offer different fertility options at different stages of the gender confirming process. Mean testicular weight was 11.35 g in our study group. Individually low testicular weight was associated with impaired spermatogenesis and low serum testosterone [20]. Hence, testicular size/weight could indicate to the physician the status of spermatogenesis and germ cell regression.

## 4. Clinical Approach towards Trans Women

Before being able to offer suitable options for fertility preservation (Figure 4), it is important to carefully obtain the medical history and perform a proper examination especially regarding the fertility status of the patient (clinical examination, blood tests, and ultrasound). The clinical examination includes examining the status of virilization and the scrotal examination to detect cryptorchidism, a tumor, or hypotrophic testes (single testis volume < 12 mL) as an indicator of impaired spermatogenesis. Additionally, the epididymis and vas deferens are palpated to exclude pathologic findings and, in particular, an absence of the vas deferens. In younger patients, it is important to determine the status of puberty using the Tanner stages. Determination of testicular volume confirms initiation of puberty at testis volumes above 4 mL. The scrotal ultrasound should be performed to substantiate the results of palpation. Regarding the epididymis, a diameter of >7.5 mm of the caput epididymis can be a hint for an obstruction proximal to the epididymis and thus a reason for impaired semen parameters. In case of epididymitis, ultrasound can reveal a thickened hyperemic epididymis. Blood tests include the determination of sex hormones (LH, FSH, testosterone, estradiol, sex-hormone binding globulin (SHBG), free testosterone, prolactin). Karyotyping should be considered if there is a clinical suspicion of an underlying genetic disorder (i.e., Klinefelter syndrome, XX male) or disorder of sexual development (DSD). Finally, an ejaculate sample can be examined to determine the actual fertility status.

## 5. Options for Fertility Preservation for Trans Women

Every trans woman should be counselled about options of fertility preservation. For those trans women who want to preserve their fertility following extensive counselling, different standardized andrological/urological procedures including ejaculate examination, cryopreservation of sperm from the ejaculate, microsurgical testicular sperm extraction (mTESE), or microsurgical epididymal sperm aspiration (MESA) can be offered during the gender confirming process in order to preserve their fertility (Figure 4). Small testicular tissues can be removed by mTESE in case of non-obstructive azoospermia in the ejaculate. This is performed under general anesthesia to isolate viable sperm for cryopreservation. An alternative approach is the MESA if complete spermatogenesis is present and there is an obstruction leading to azoospermia in the ejaculate. In MESA, sperm-containing fluid is retrieved from the epididymis through needle aspiration.

Cryopreservation of immature testicular tissues containing spermatogonia as most differentiated germ cell type is already offered to pre-pubertal and early pubertal patients prior to initiation of spermatogenesis and at risk of germ cell loss (i.e., oncological patients) in Germany (ANDROPROTECT^®^) as well as in other European centers [1,49]. The experimental procedures of extracting and cryopreserving immature testicular tissues containing SSCs can also be an option to preserve fertility in pre-pubertal trans women or in patients whose spermatogenesis is already regressed and only contains undifferentiated germ cells. However, this approach is still considered experimental.

### 5.1. Fertility Preservation Options before Initiation of GCHT

If patients present normozoospermia or oligoasthenoteratozoospermia syndrome (OAT-syndrome) after the diagnosis and before starting GCHT, cryopreservation of sperm after masturbation can be offered. If masturbation is emotionally challenging [8] or the patient is azoospermic because of individual comorbidities (i.e., maldescensus testis), mTESE or MESA can be pursued. The success rate depends on the quality and quantity of sperm that can be cryopreserved [24]. Regarding fertility preservation, sperm can be cryopreserved at different centers in Germany. An annual fee is paid for cryostorage. However, health policies are currently being reconsidered and fertility protection might be free of charge for German infertility patients including gender dysphoria patients in the future.

### 5.2. Fertility Preservation Options during GCHT

If patients have already started GCHT, but blood values, testicular size, ultrasound, and ejaculate examinations still suggest the presence of intact spermatogenesis, cryopreservation of ejaculated sperm could be offered after masturbation. In azoospermic patients or emotionally challenged patients, mTESE/MESA or experimental methods of extracting and cryopreserving testis tissue only containing spermatogonia can be discussed.

### 5.3. Fertility Preservation on the Day of GCS

If clinicians fail to discuss fertility preservation with the patients or if patients are emotionally challenged by masturbation or financially challenged by an extra procedure/operation during the gender confirming process, (m)TESE methods could still be performed on the day of GCS, looking for spermatozoa during or after the removal of testes. Furthermore, cryopreservation of testicular tissues containing spermatogonia could be offered on the day of GCS in centers where protocols are established and ethical approval is granted for clinical studies.

### 5.4. Fertility Preservation Options after GCS

It was shown that up to 80% of transgender persons do have sexual relationships during the gender confirming process and after GCS [50]. If trans women have cryopreserved sperm or testicular tissue and are in a relationship with a woman who is still able to reproduce, standard methods in reproductive health clinics (i.e., in-vitro fertilization, intracytoplasmic sperm injection) can be offered [24]. If trans women stay single or are in a relationship with men, surrogacy (+/− oocyte donation) can be pursued in some countries. However, the latter is not allowed in Germany.

Our Centre of Reproductive Medicine and Andrology includes a Department of Clinical and Surgical Andrology. Fertility counselling and fertility preservation are among the services offered to male patients in the clinic. Between December 2012 and February 2019, 16 trans women were counselled for fertility preservation options. Five of these patients (aged between 13 and 62 years) successfully cryopreserved semen specimens with terato- and normozoospermia. In addition to that, two individuals (22 and 33 years) underwent TESE due to azoospermia. To date, none of the individuals who have cryopreserved semen have used these samples in the frame of an assisted reproductive treatment.

## 6. Conclusions

This review provides and overview regarding the fertility preservation options for trans women. It is of note, however, that trans men and non-binary people may have different needs, which are not addressed in this review. It is advised to discuss fertility options as early as possible during the gender confirming process with trans women in Germany. At this stage of treatment, trans women have intact testes with complete spermatogenesis. During and after GCHT, trans women have disturbed or absent spermatogenesis due to the negative central effect of the hormonal regimens on spermatogenesis. Counselling early increases the chances to retrieve sperm and therefore offers cheap and less-invasive methods for fertility preservation.

National and international guidelines demand to counsel trans women about fertility options as early as possible. However, for many reasons, only a minority of trans women undertake efforts to preserve their fertility. To conclude, we strongly advise to counsel trans women regarding fertility preservation options and discuss individual options as early as possible in the gender confirming process in order to fulfill the individual reproductive wish to have one’s own biological children.

## Figures and Tables

**Figure 1 jcm-08-00730-f001:**
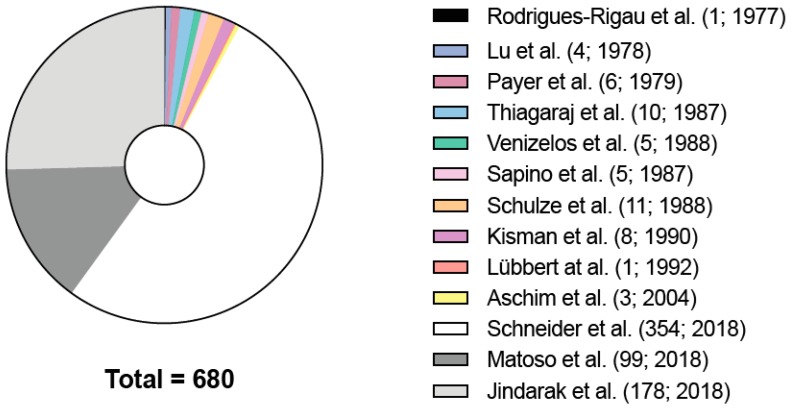
Studies on testicular histology of transgender patients. The 13 publications that analyzed testicular histology (*n* = 680) of transgender patients. In brackets are the total numbers of testicular tissues assessed in the respective publications. Rodrigues-Rigau et al. (1; 1977) [37], Lu et al. (4; 1978) [38], Payer et al. (6, 1979) [39], Thiagaraj et al. (10; 1987) [40], Venizelos et al. (5; 1987) [41], Sapino et al. (5; 1987) [42], Schulze et al. (11; 1988) [43], Kisman et al. (8; 1990) [44], Lübbert et al. (1; 1992) [45], Aschim et al. (3; 2004) [46], Matoso et al. (99; 2018) [21], Jindarak et al. (178; 2018) [22], Schneider et al. is an update of a previous study.

**Figure 2 jcm-08-00730-f002:**
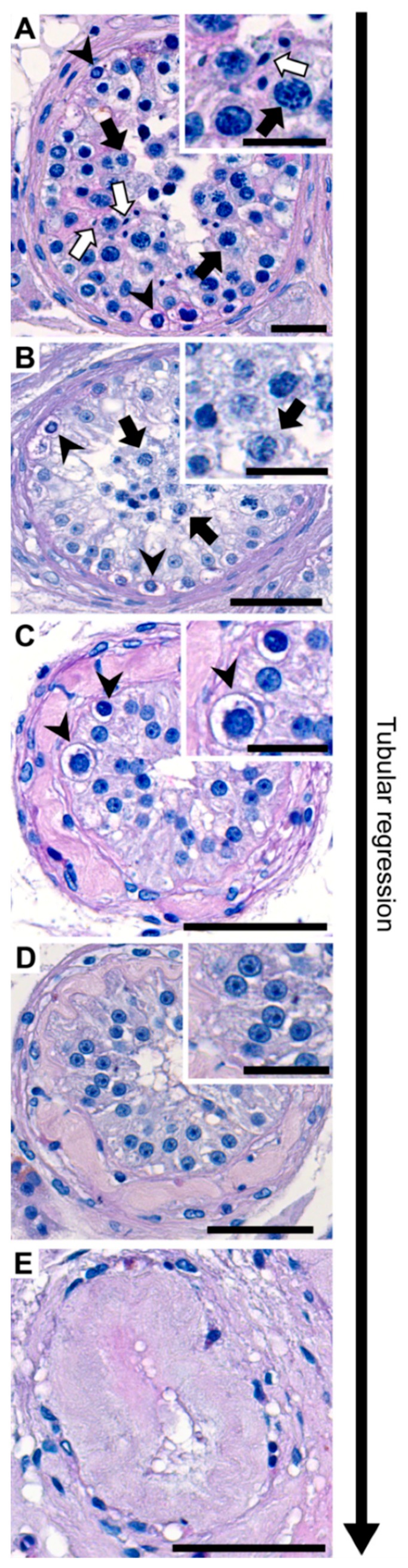
Histological findings showing stages of tubular regression in testicular tissues of transgender patients. Representative periodic acid-Schiff stained seminiferous tubules. (**A**) Tubule with complete spermatogenesis contains different types of germ cells. Spermatogonia (arrowhead), spermatocytes (black arrow) and elongated spermatids (white arrow). (**B**) In meiotic arrest spermatogonia (arrowhead) and spermatocytes (black arrow) are present. (**C**) Spermatogonial arrest shows only spermatogonia (arrowhead). (**D**) Sertoli cell only phenotype without germ cells. (**E**) Tubular shadow without any cells in the lumen. Tubular regression is described from A–E.

**Figure 3 jcm-08-00730-f003:**
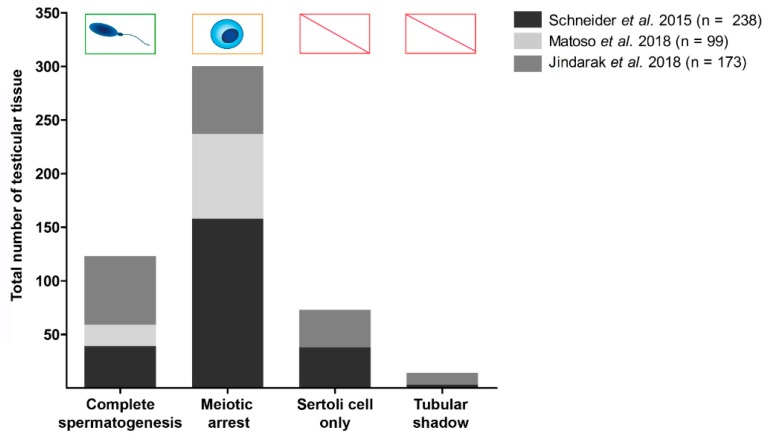
State of spermatogenesis of the testicular tissues of the transgender patients. Total numbers of testicular tissues with a specific state of spermatogenesis are shown in a cohort of 510 patients. Dark grey bars represent 238 testicular tissues [24], grey bars the results of 173 testicular tissues, and light grey bars the results of 99 testicular tissues [26]. The chance to isolate sperm and spermatogonia from respective tissues is indicated above the bars. Abbreviations stand for: CS: Complete spermatogenesis, MA: Maturation arrest, SCO: Sertoli cell only and TS: Tubular shadow.

**Figure 4 jcm-08-00730-f004:**
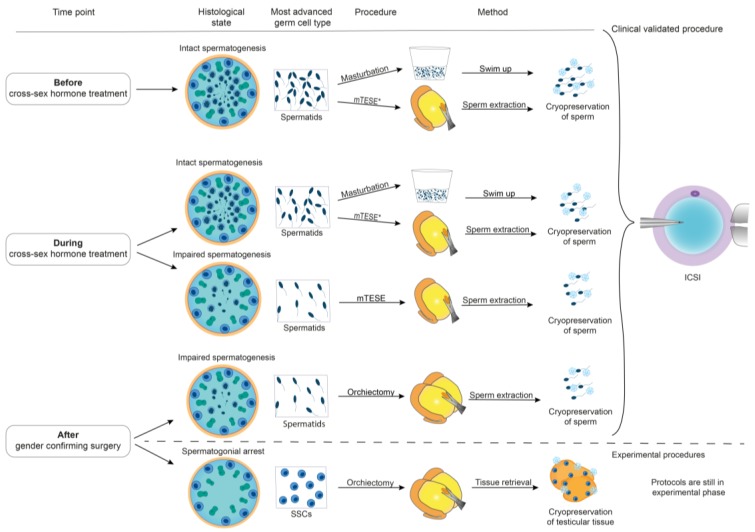
Different options for fertility preservation depending on the time point of Gender confirming hormone therapy (GCHT) and spermatogenic status. During the transition, the histological state of the testes shows a regression of spermatogenesis to severely impaired spermatogenesis or even spermatogonial arrest. Depending on the most advanced germ cell type, different procedures can be offered to preserve germ cells. Before and during cross-sex hormone treatment, masturbation is the most favorable non-invasive option. Following masturbation, semen samples are processed by swim up and sperm are cryopreserved. Alternatively, microsurgical testicular sperm extraction (mTESE) can be performed. After mTESE, sperm need to be extracted prior to cryopreservation, and the same applies for sperm extraction after orchiectomy. Cryopreserved sperm can be used for ICSI (intracytoplasmic sperm injection). Testicular tissues with spermatogonial stem cells (SSC) can also be cryopreserved although this procedure is still experimental. Abbreviations are: mTESE: Microsurgical testicular sperm extraction, ICSI: Intracytoplasmic sperm injection, SSCs: Spermatogonial stem cells. *mTESE can be offered if masturbation is psychologically distressing.

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
