# Peer review of "Options for Fertility Treatments for Trans Women in Germany"

_jcm, 2019, doi:10.3390/jcm8050730_

Round 1
Reviewer 1 Report
Thank you for the opportunity to review this paper on such an important topic in trangender care. The purpose of the following comments is the help strengthen this important paper:
1. Please use consistent terminology throughout the paper, including the title of the paper. Consider using the transwomen, MtF transgender persons, or a similar label as opposed to MtF GD patients so that the terminology used is not diagnosis focused).
2. Page 2, Line 51: The sentence “A decisive step is the gender confirming surgery (GCS)” seems to be abruptly placed. It is unclear the purpose of this sentence at this place in the paragraph. Consider adding more text to contextualize this sentence. Another option would be to delete this sentence.
3. In the introduction it would be helpful for the authors to explain why it is important to have discussions about fertility preservation with patients who are considering gender affirming medical interventions and provide some examples for readers who are not familiar. There is some information already provided that can serve as examples if some of the text were restructured.
4. Page 2, Line 67: In the sentence that starts, “CSHT consists of treatment with anti-androgens… “, considering adding CSHT for transwomen (or MtF GD persons to be consistent with the language that authors have chosen). This small additional with minimize confusion for readers who are not as familiar with hormone therapies.
5. Line 77: For the section that reads “fertility loss was not relevant to defer their transition process”, do the authors mean that fertility loss was not relevant to defer decision to engage in gender affirming medical interventions? If so, I would clarify this point.
6. Page 2, Line 78: In some parts of the world, transsexual is a word that is avoided due to it being considered pejorative. Consider using a different word.
7. For the paragraph starting on page 3, line 116, please add a header to orient readers. This paragraph seems to no longer be a part of the introduction and instead seems to transition into the methods.
8. Please include information about consent procedures for patients who completed the questionnaires and procedures for testicular tissue and blood serum collection.
9. Was there any ethical board reviews for this project? Please include this information, if relevant.
10. Page 4, line 136: please consider using the language 99 testes from transgender individuals as opposed to “99 transgender testes”
11. Page 6, line 179 – please consider using different terminology than “non-transsexual men” (please see comment #5 above)
12. In the section called, “Options for fertility preservation for GD patients”, consider adding some text to explain that not all transgender people or people with GD are interested in fertility preservation but it is important to discuss this with patients of all ages (who are interested in gender affirming medical interventions that might compromise fertility).
13. Page 9, line 45 – please remove the word “homosexual”. Simplying saying “MtF persons… in a relationship with a woman” is adequate. Transwomen may or may not use the term homosexual to describe their relationships, attractions, etc. Also, this is a term that is avoided in some part of the world.
14. Page 9, line 47 – Consider changing the wording of this sentence to, “If MtF persons stay single or are in relationships with men…”
Author Response
Reviewer 1
The authors provide a useful summary of specific techniques, and i appreciated their strong advocacy for the rights of transgender women to fertility treatment. My enthusiasm for the paper, however, was tempered by some problems with language, as outlined below, and a lack of summary of previous literature.
Response: We thank the reviewer for the appreciation of our work. We have addressed all issues raised point by point and provide a more comprehensive literature review in the revised version.
1) I think the language used is not appropriate. Male to female (or MtF) is very outdated language. Gender dysphoria persons has little to no meaning. I would suggest using transfeminine people or transgender women or similar.
Response: Following this reviewer’s advice, we have changed “gender dysphoria” to “transgender”. We have changed “Male to Female” to “transwoman” throughout the manuscript.
2) The abstract is incorrect, in that fertility treatment is widely available in many countries. If you mean specifically in Germany it is hard to access, then you should say that.
Response: We thank the reviewer for pointing out this important issue: We have modified the abstract as suggested (lines 25 ff.): Fertility preservation in transwomen is a crucial but thus far neglected component in the gender confirming treatment in Germany. It is difficult for transwomen to access reproductive health care, because centers offering treatment, psychological guidance, gender confirming surgery as well as reproductive health services are scarce in Germany.
3) Again, 'gender dysphoria patients' has little meaning, and could be read as pathologising. I would just use 'transgender people' or if you are speaking about transgender women specifically, then say that.
Response: Thank you very much for your comment. We have changed “gender dysphoria” to transgender people.
4) There is now research on children conceived by transgender people using ARTs, please cite (eg studies of transgender men who have undertaken a pregnancy).
Response: In line with the reviewer´s suggestion we have reviewed suggested literature and we haved added it in ll. 63-66: Limited information is available on the welfare of children born to transgender persons and conceived via assisted reproduction [5]. However, a recent study reported no pathalogies in well-being or psychological development and no difficulties in their relationships with their parents detected [6].
Additionally we have added in (ll. 81-86): The majority (70 %) of German transgender patients wish to have children [16]. In Germany, two out of 99 transwomen and three out of 90 transmen had their own children, but four of these five individuals stated not to have used their own oozytes or sperm [16]. Rather, 16 % of transmen had biological children with donor sperm [17]. Apart from that, 88% of transmen used their own oocytes and mean age at conception was 28 [18].
5) There is one very short paragraph on fertility treatment by transgender people. There has been a LOT of research on this topic in the past 2 years, which should be cited and summarised.
Response: Thank you very much for your comment. We included in the introduction the following sentences:
Lines 72 ff.: The majority of patients of a Belgian study would have considered sperm freezing if it had been presented as an option [8] and only a small percentage thought that sperm freezing was in conflict with their female identity [15].
Additionally we included the suggested publications in the introduction:
Lines 74 ff.: Reports of transgender individuals seeking counselling is increasing worldwide [9, 10, 11, 12, 13], however the proportion of patients of the published cohort taking measures to preserve fertility is still low. Moreover, interest in fertility preservation among transgender individuals may highly differ in different countries [14].
6) There is just one small paragraph on outcomes from the authors' clinic data. I would have appreciated some more information about the sample and clinic setting.
Response: We thank the reviewer for the opportunity to provide more information regarding the clinical setting. We have modified the pragraph as follows (ll. 317-325): Our Centre of Reproductive Medicine and Andrology includes a Department of Clnical and Surgical Andrology. Fertility counselling and fertility preservation are among the serviced offered to male patients in the clinic. Between 12/2012 and 02/2019, 16 transwomen were counselled for fertility preservation options between 12/2012 till 02/2019. Five of these patients (aged between 13-62 years) successfully cryopreserved semen specimens with terato- and normozoospermia. In addition to that, two individuals (22 and 33 years) underwent TESE due to azoospermia. To date, none of the individuals who have cryopreserved semen have used these samples in the frame of an assisted reproductive treatment.
7) The conclusion should acknowledge again that the paper only focuses on transgender women/girls, and that other transgender and non-binary people will have different needs.
Response: We absolutely agree with this comment and we have revised the conclusion and highlighted that this manuscript describes the possiblities for transwomen.
We included the following paragraph in ll. 328-330:
This review provides and overview regarding the fertility preservation options for transwomen. It is of note, however, that transmen and non-binary people may have different needs, which are not addressed in this review.
Reviewer 2 Report
The authors provide a useful summary of specific techniques, and i appreciated their strong advocacy for the rights of transgender women to fertility treatment. My enthusiasm for the paper, however, was tempered by some problems with language, as outlined below, and a lack of summary of previous literature.
1) I think the language used is not appropriate. Male to female (or MtF) is very outdated language. Gender dysphoria persons has little to no meaning. I would suggest using transfeminine people or transgender women or similar.
2) The abstract is incorrect, in that fertility treatment is widely available in many countries. If you mean specifically in Germany it is hard to access, then you should say that.
3) Again, 'gender dysphoria patients' has little meaning, and could be read as pathologising. I would just use 'transgender people' or if you are speaking about transgender women specifically, then say that.
4) There is now research on children conceived by transgender people using ARTs, please cite (eg studies of transgender men who have undertaken a pregnancy)
5) There is one very short paragraph on fertility treatment by transgender people. There has been a LOT of research on this topic in the past 2 years, which should be cited and summarised. See Riggs, D.W., & Bartholomaeus, C. (2018). Fertility preservation decision making amongst Australian transgender and non-binary adults. Reproductive Health, 15, 181-191 for a summary.
6) There is just one small paragraph on outcomes from the authors' clinic data. I would have appreciated some more information about the sample and clinic setting.
7) The conclusion should acknowledge again that the paper only focuses on transgender women/girls, and that other transgender and non-binary people will have different needs.
Author Response
Reviewer 2
This review article entitled "Options for fertility treatments for male to female gender dysphoria persons in Germany" states the objective of providing an overview on options for fertility preservation in patients with gender dysphoria. The authors begin by detailing the importance of discussing fertility preservation early on in the hormonal treatment of individuals with gender dysphoria, and describe some reports in the literature which show that fertility can be preserved in male to female individuals who have already begun gender affirming hormone treatment. They then discuss a variety of issues, including the availability of gender preservation options in Germany, testicular steroidogenesis, a study performed at their institution, and end with recommendations for fertility preservation in this patient population.
While the issue of fertility preservation in patients with gender dysphoria is certainly important, this paper has several major issues that raise potential concern.
This manuscript is presented as a review, however in addition to some literature review, which is not comprehensive, the authors allude to their own, presumedly unpublished research study in several areas of the paper (lines 116-132, 146-151, page 9 lines 50-55). If the authors are presenting unpublished research, this would be better presented in an original research manuscript, rather than mentioned in passing in a review article. Additionally, the study is referred to in several different parts of the manuscript, but is never described fully, and is lacking specific details of the methods and complete results.
Response: We fully appreciate the concern of this reviewer. In 2015 we published results based on a smaller cohort of patients (n = 108). Characterization of samples has been continued since the employing the published methodology. We have clarified this in the respective positions of the manuscript (ll. 164 ff).
There are some major organizational flaws, including the fact that after the lengthy introduction which does not mention anything regarding the potential mechanisms for decreased fertility in the selected patient population, the authors only briefly describe steroidogenesis, then begin referring to their study as detailed above (lines 116-132). Line 133 begins an entirely different theme, focusing on other reports of testicular histology findings in patients who have been treated with GAHT, and then in line 146 the authors again discuss their study.
Response: Thank you very much for your comment. In the introduction we describe the mechanisms of gender confirming hormone therapy: GCHT in transwomen consists of treatment with anti-androgens (Androcur®) or spironolactone or gonadotropin releasing hormone agonists in combination with estrogens (oral, transdermal, intramuscular). This leads to down-regulation of gonadotropin releasing hormone, followed by a prominent decline of luteinizing hormone (LH) and follicle stimulating hormone (FSH). This leads to testicular regression associated with low levels of serum androgens and suppression of spermatogenesis. After several weeks GCHT interrupts germ cell differentiation at the pre-meiotic stage [20,21,22].
Figure 1 states that it refers to studies on testicular histology of patients, however there is no useful information provided by the figure, other than the references for the studies.
Response: We do not fully agree with this comment. We have included this figure to provide an overview on the numbers of testicular tissues that have been examined/published. This figure highlights the latest three studies which included comparatively higher number compared to previous publications.
Figure 3 combines 3 different studies into one, and references the chance to isolate genetic material from tissues, however this information does not appear to be included in the figure itself.
Response: We thank the reviewer for this comment and have revised the figure to improve clarity. The revised image includes a more detailed description of the information provided on the y-axis. Moreover, the chances to isolate sperm and spermatogonia, respectively, is indicated by the same images used in figure 4, serving as a link between the figures. Finally, the fact that solely sperm can currently be used in the frame of assisted reproductive treatments is indicated by the box displayed above the respective bar graph.
The paragraph beginning with line 181 seems out of place after a sentence detailing immunohistochemical markers, and would be better served as its own section.
Response: In line with this comment, we linked both paragraphs (ll. 219 – 222): To assess characteristics of cell types in testicular tissues of transwomen immunohistochemical stainings were performed. The use of immunohistochemical markers (LHCGR, SMA, VIM, SALL 4, MAGE A4, VASA) revealed that the expression patterns in transwomen were identical and in agreement with the staining pattern in samples from non-transgender men [38]. This finding was confirmed by others [21, 22].
On page 8 (line number resets) beginning with line 5, MESA is referred to, however this process is not described in the paper. The authors would benefit from pending more time describing potential fertility techniques that they reference.
Response: To address this point, we have added the following sentences: Small testicular tissues can be removed by microsurgical testicular sperm extraction (mTESE) in case of non-obstructive azoospermia in the ejaculate. This is performed under general anesthesia to isolate viable sperm for cryopreservation. An alternative approach is the microsurgical epididymal sperm extraction (MESA), if complete spermatogenesis is present and there is an obstruction leading to azoospermia in the ejaculate. In MESA sperm-containing fluid is retrieved from the epididymis through needle aspiration.
The overall flow of the paper does not contribute to a conducive, logical read. There are also multiple terminology errors, including referred to individuals with gender dysphoria as "gender dysphoric individuals," which is a less-accepted term in the transgender community. The authors also use terms used such as "homosexual," "transexual," and ""cross-sex hormone therapy," which are not currently preferred, and "maldescensus testis," rather than undescended testicle or cryptorchidism, both of which are more commonly used in the medical literature.
Response: We have changed “gender dysphoria” to “transgender”, “Male to Female” to “transwomen” and “cross-sex hormone therapy” to “gender confirming hormone therapy”.
The paper also contains grammatical errors, capitalization inconsistencies, and several brand names for medications and procedures, which ideally should be avoided.
Response: We have revised the manuscript extensively to improve the quality of the grammar.
Reviewer 3 Report
This review article entitled "Options for fertility treatments for male to female gender dysphoria persons in Germany" states the objective of providing an overview on options for fertility preservation in patients with gender dysphoria. The authors begin by detailing the importance of discussing fertility preservation early on in the hormonal treatment of individuals with gender dysphoria, and describe some reports in the literature which show that fertility can be preserved in male to female individuals who have already begun gender affirming hormone treatment. They then discuss a variety of issues, including the availability of gender preservation options in Germany, testicular steroidogenesis, a study performed at their institution, and end with recommendations for fertility preservation in this patient population.
While the issue of fertility preservation in patients with gender dysphoria is certainly important, this paper has several major issues that raise potential concern.
This manuscript is presented as a review, however in addition to some literature review, which is not comprehensive, the authors allude to their own, presumedly unpublished research study in several areas of the paper (lines 116-132, 146-151, page 9 lines 50-55). If the authors are presenting unpublished research, this would be better presented in an original research manuscript, rather than mentioned in passing in a review article. Additionally, the study is referred to in several different parts of the manuscript, but is never described fully, and is lacking specific details of the methods and complete results.
There are some major organizational flaws, including the fact that after the lengthy introduction which does not mention anything regarding the potential mechanisms for decreased fertility in the selected patient population, the authors only briefly describe steroidogenesis, then begin referring to their study as detailed above (lines 116-132). Line 133 begins an entirely different theme, focusing on other reports of testicular histology findings in patients who have been treated with GAHT, and then in line 146 the authors again discuss their study.
Figure 1 states that it refers to studies on testicular histology of patients, however there is no useful information provided by the figure, other than the references for the studies.
Figure 3 combines 3 different studies into one, and references the chance to isolate genetic material from tissues, however this information does not appear to be included in the figure itself.
The paragraph beginning with line 181 seems out of place after a sentence detailing immunohistochemical markers, and would be better served as its own section.
On page 8 (line number resets) beginning with line 5, MESA is referred to, however this process is not described in the paper. The authors would benefit from pending more time describing potential fertility techniques that they reference.
The overall flow of the paper does not contribute to a conducive, logical read. There are also multiple terminology errors, including referred to individuals with gender dysphoria as "gender dysphoric individuals," which is a less-accepted term in the transgender community. The authors also use terms used such as "homosexual," "transexual," and ""cross-sex hormone therapy," which are not currently preferred, and "maldescensus testis," rather than undescended testicle or cryptorchidism, both of which are more commonly used in the medical literature. The paper also contains grammatical errors, capitalization inconsistencies, and several brand names for medications and procedures, which ideally should be avoided.
Author Response
Reviewer 3
Thank you for the opportunity to review this paper on such an important topic in trangender care. The purpose of the following comments is the help strengthen this important paper:
Response: We greatly appreciate the appraisal of our work.
1. Please use consistent terminology throughout the paper, including the title of the paper. Consider using the transwomen, MtF transgender persons, or a similar label as opposed to MtF GD patients so that the terminology used is not diagnosis focused).
Response: In line with this comment we have changed “gender dysphoria” to transgender; “Male to Female” to transwoman and “cross-sex hormone therapy” to “gender confirming hormone therapy”.
2. Page 2, Line 51: The sentence “A decisive step is the gender confirming surgery (GCS)” seems to be abruptly placed. It is unclear the purpose of this sentence at this place in the paragraph. Consider adding more text to contextualize this sentence. Another option would be to delete this sentence.
Response: We have deleted this sentence.
3. In the introduction it would be helpful for the authors to explain why it is important to have discussions about fertility preservation with patients who are considering gender affirming medical interventions and provide some examples for readers who are not familiar. There is some information already provided that can serve as examples if some of the text were restructured.
Thank you for your comment. We have added the following paragraph to the introduction (ll. 103 ff): Different standardized andrological/urological procedures including cryopreservation of sperm from the ejaculate, microsurgical testicular sperm extraction (mTESE) or microsurgical epididymal sperm aspiration (MESA) can be offered during the gender confirming process in order to preserve fertility in transwomen. Counselling as early as possible in the gender confirming process is crucial as the testes are still intact providing the chance to cryopreserve sperm from the ejaculate. Cryopreservation of spermatogonial stem cells is also possible, however using them for fertility procedures is not yet an established clinical routine but still in the experimental phase.
4. Page 2, Line 67: In the sentence that starts, “CSHT consists of treatment with anti-androgens… “, considering adding CSHT for transwomen (or MtF GD persons to be consistent with the language that authors have chosen). This small additional with minimize confusion for readers who are not as familiar with hormone therapies.
Response: In line with this comment, we have included “in transwomen” in the respective sentence.
5. Line 77: For the section that reads “fertility loss was not relevant to defer their transition process”, do the authors mean that fertility loss was not relevant to defer decision to engage in gender affirming medical interventions? If so, I would clarify this point.
Response: As suggested we have revised the sentence (ll. 114 ff): 90% of transgender persons stated that fertility loss was not relevant to defer the decision to engage in gender affirming medical interventions.
6. Page 2, Line 78: In some parts of the world, transsexual is a word that is avoided due to it being considered pejorative. Consider using a different word.
Response: Thank you very much for your valuable comment: We have replaced the word “transsexual” by “transgender” throughout the manuscript.
7. For the paragraph starting on page 3, line 116, please add a header to orient readers. This paragraph seems to no longer be a part of the introduction and instead seems to transition into the methods.
Response: Following the reviewers suggestions we have added the following heading to the respective paragraph (l. 162): Testicular tissues and steroidogenesis in our cohort of transwomen
8. Please include information about consent procedures for patients who completed the questionnaires and procedures for testicular tissue and blood serum collection.
Response: We thank the reviewer for bringing up this important point: We have included the following sentence (ll. 168 ff):
Ethical approval was received from the Ethical Committee of the Ärztekammer Westfalen-Lippe (no. 2012-555-f-S) and the Ärztekammer Hamburg (no. MC-131/13) prior to study initiation and written informed consent was obtained from each subject prior to study participation.
9. Was there any ethical board reviews for this project? Please include this information, if relevant.
Response: Please refer to the previous point.
10. Page 4, line 136: please consider using the language 99 testes from transgender individuals as opposed to “99 transgender testes”
Response: We have revised the sentence as suggested.
11. Page 6, line 179 – please consider using different terminology than “non-transsexual men” (please see comment #5 above)
Response: In line with this comment we have replaced “non-transsexual” by “non-transgender” throughout the manuscript.
12. In the section called, “Options for fertility preservation for GD patients”, consider adding some text to explain that not all transgender people or people with GD are interested in fertility preservation but it is important to discuss this with patients of all ages (who are interested in gender affirming medical interventions that might compromise fertility).
Response: Thank you very much for your comment. We added the following sentence as an introduction to the paragraph (ll. 256 ff): Every transwomen should be counselled about options of fertility preservation. For those transwomen, who want to preserve their fertility following extensive counselling, different standardized andrological/urological procedures including ejaculate examination, cryopreservation of sperm from the ejaculate, microsurgical testicular sperm extraction (mTESE) or microsurgical epididymal sperm aspiration (MESA) can be offered during the gender confirming process in order to preserve fertility in transwomen.
13. Page 9, line 45 – please remove the word “homosexual”. Simplying saying “MtF persons… in a relationship with a woman” is adequate. Transwomen may or may not use the term homosexual to describe their relationships, attractions, etc. Also, this is a term that is avoided in some part of the world.
Response: We appreciate this comment and have removed the words “homosexual” and “heterosexual” from this paragraph.
14. Page 9, line 47 – Consider changing the wording of this sentence to, “If MtF persons stay single or are in relationships with men…”
Response: We have revised the sentence as suggested and have removed the words “homosexual” and “heterosexual” in this paragraph.
Round 2
Reviewer 2 Report
The authors have done an excellent job of revising the paper as requested. One terminological issue remains: writing 'transwomen' is like writing 'lesbianwomen' or 'gaymen'. Running the two words together means that 'trans' is no longer used as an adjective, and instead 'transwomen' are treated as a separate group of humans entirely from women. The authors should change throughout to 'trans women'.